# Treatment Outcomes of Novel Targeted Agents in Relapse/Refractory Chronic Lymphocytic Leukemia: A Systematic Review and Network Meta-Analysis

**DOI:** 10.3390/jcm8050737

**Published:** 2019-05-23

**Authors:** Po-Huang Chen, Ching-Liang Ho, Chin Lin, Yi-Ying Wu, Tzu-Chuan Huang, Yu-Kang Tu, Cho-Hao Lee

**Affiliations:** 1Division of Hematology and Oncology Medicine, Department of Internal Medicine, Tri-Service General Hospital, National Defense Medical Center, Taipei 114, Taiwan; chenpohuang@hotmail.com (P.-H.C.); charileho22623@gmail.com (C.-L.H.); doc10456@mail.ndmctsgh.edu.tw (Y.-Y.W.); tzuchuanhuang@mac.com (T.-C.H.); 2School of Public Health, National Defense Medical Center, Taipei 114, Taiwan; xup6fup0629@gmail.com; 3Department of Research and Development, National Defense Medical Center, Taipei 114, Taiwan; 4Institute of Epidemiology and Preventive Medicine, College of Public Health, National Taiwan University, Taipei 100, Taiwan; 5Department of Medical Research, National Taiwan University Hospital, National Taiwan University, Taipei 100, Taiwan

**Keywords:** targeted agents, kinase inhibitors, small molecule inhibitors, relapse or refractory chronic lymphoblastic leukemia, network meta-analysis

## Abstract

Most chronic lymphocytic leukemia patients experience a relapse or become refractory to treatment with conventional chemotherapeutic agents. The network meta-analysis assesses the relative efficacy of novel targeted agents for the treatment of a relapse or refractory chronic lymphocytic leukemia. A systematic literature search included seven phase III randomized controlled trials, including a total of 2512 patients treated with nine regimens. Data were extracted and evidence synthesized using network meta-analysis. All novel targeted therapies were significantly more effective than ofatumumab and demonstrated promising prolongation of progression free survival (PFS), with a hazard ratio (HR) ranging from 0.10 to 0.52. Two novel targeted agent regimens, venetoclax plus rituximab and ibrutinib monotherapy, resulted in greater overall survival (HR, 0.335 and 0.361, respectively). Venetoclax plus rituximab and ibrutinib monotherapy were most favorable based on (1) HR for PFS compared with ofatumumab (Ibrutinib: HR, 0.10; 95% CI, 0.07–0.14; Venetoclax plus rituximab: HR, 0.10; 95% CI, 0.05–0.21) and SUCRA value (probability of being best) (Ibrutinib SUCRA, 0.92; Venetoclax rituximab SUCRA, 0.90) (2) HR for overall survival compared with ofatumumab (Ibrutinib: HR, 0.361; 95% CI, 0.208–0.627; Venetoclax rituximab: HR, 0.335; 95% CI, 0.112–0.997) and SUCRA value (Ibrutinib SUCRA, 0.84; Venetoclax rituximab SUCRA, 0.85) Both treatments reduced the risk of progression or death by 90% versus conventional ofatumumab. Both ibrutinib monotherapy and venetoclax rituximab have a high probability of being the most effective treatments for a relapse or refractory chronic lymphocytic leukemia with respect to long-term progression-free survival and overall survival.

## 1. Introduction

Chronic lymphocytic leukemia (CLL) is the most common B-cell malignancy in adults of the Western world. An annual incidence of two to six per 100,000 people has been reported, with a median age of 72 years at initial diagnosis [1]. The clinical course of CLL is heterogeneous. Most patients are diagnosed with early stage, indolent disease. However, in some high-risk patients, particularly those with gene mutations, such as the 17p deletion, unmutated IGHV (Immunoglobulin heavy variable region), and 11q deletion, the disease may develop rapidly. These patients require further treatment for disease progression [2].

Chemoimmunotherapy, known as the FCR regimen (Fludarabine, Cyclophosphamide. and Rituximab), now prolongs the overall survival (OS) of CLL patients and as such is the recommended first-line treatment for patients without the 17p deletion [3]. However, CLL remains incurable and eventually relapses or becomes refractory to first-line treatment. Many of the patients are elderly, with toxicity and comorbidities resulting from aggressive chemoimmunotherapy. As a result, treating these patients is a challenge.

Traditional therapy for relapsed or refractory chronic lymphocytic leukemia (R/R CLL) has included alkylating agents, such as bendamustine, and anti-CD20 monoclonal antibodies, such as rituximab and ofatumumab. A commonly used regimen has been bendamustine in combination with rituximab or single-agent ofatumumab or single-agent rituximab. Recently, a range of novel targeted agents, including ibrutinib, venetoclax, idelalisib, and duvelisib, have provided greater survival with a modest toxicity profile. These new treatments were established by pivotal randomized controlled trials (RCTs) in patients with R/R CLL [4,5,6,7,8,9,10,11]. Ibrutinib is a once-daily, oral covalent inhibitor of Bruton’s tyrosine kinase, which inhibits B-cell receptor signaling [12]. Venetoclax is an orally administered potent and selective BH3 mimetic that targets the BCL-2 (B-Cell Leukemia/Lymphoma 2 Gene) inhibitor, exhibiting significant apoptotic activity [13]. Idelalisib is an orally bioavailable selective small-molecule inhibitor of phosphatidylinositol 3-kinase δ (PI3Kδ) [14]. Duvelisib is an oral dual inhibitor of PI3K-δ,γ [15]. 

The results of clinical trials evaluating progression-free survival (PFS) and OS of small-molecule inhibitors, with or without other anti-leukemic drugs, have been reported. The outcomes of most of these treatments have been evaluated by–comparisons to previous traditional regimens, either bendamustine combined with rituximab or ofatumumab or rituximab monotherapy. However, no study has presented a head-to-head comparison of the efficacy of these new small-molecule inhibitors in treating R/R CLL patients. We conducted this network meta-analysis to assess the relative efficacy of each drug with the aim of providing treatment recommendations to physicians in daily clinical practice.

## 2. Materials and Methods

### 2.1. Search Strategy

An electronic search for relevant publications was performed using PubMed, Embase, LILACS database, the Cochrane collaboration database, and proceedings from major international meetings in hematology and oncology, such as the American Society of Clinical Oncology (ASCO 2018) and European Hematology Association (EHA 2018). Comprehensive search strategies were used to identify all relevant trials (Appendix A). All titles were screened, and abstracts were reviewed. To ensure that no RCTs were missing, the related article references were reviewed, and searches were conducted using Google Scholar, the US government clinical trials database (www.ClinicalTrials.gov) and the UK National Research Register.

Articles were searched for publication dates ranging from 1 January 2005 to 6 January 2019. Studies were included if patients with R/R CLL were randomly allocated to therapies involving novel targeted agents or conventional regimens. In addition, the reported clinical outcomes had to include either progression-free survival (PFS) or overall survival (OS). The primary outcome of our study was PFS, defined as the time interval from randomization to definitive disease progression, relapse or death from any cause. The secondary outcome was OS, defined as the time interval from randomization to death from any cause.

This systematic review is registered with PROSPERO (CRD42018088179) and was performed according to the guidelines of the Preferred Reporting Items for Systematic Reviews and Meta-Analyses (PRISMA) extension statement for network meta-analysis [16,17].

### 2.2. Inclusion Criteria

To be included in the analysis, the studies had to meet the following criteria: (1) Patient cohort composed only of patients with a diagnosis of R/R CLL (according to the International Workshop on Chronic Lymphocytic Leukemia criteria) [18], (2) all patients in the cohort had relapsed or refractory diseases following one or more previous lines of systemic therapy, (3) randomized phase III clinical trial, with or without blinding, (4) abstracts or unpublished data were included only if sufficient information regarding study design, participant characteristics, interventions, and outcomes was provided.

### 2.3. Exclusion Criteria

Studies were excluded if they were not phase III trials, not comparative, not RCT (must include historical studies, dose-escalation analysis, or safety evaluation analysis), if outcomes of interest (PFS and OS) were not reported, if the methodology was not clearly reported, and if the intervention treatment did not include novel targeted agents. As this study aims to analyze patients with R/R CLL, we excluded clinical trials that included treatment-naïve patients. 

### 2.4. Risk of Bias Assessment

Study quality was evaluated by three reviewers (P.-H.C., C.-H.L., and C.L.) using the methodology and categories described in the Cochrane Collaboration Handbook [19]. In case of disagreement, a group discussion was conducted to reach a consensus. Other issues considered included baseline imbalance and the source of financial support [20]. We present risk of bias graphs created using Review Manager 5.3 software (The Cochrane Collaboration, Copenhagen, Denmark) [21].

### 2.5. Data Extraction

Two reviewers (P.-H.C. and C.-H.L.) independently assessed the eligibility of all identified citations and extracted data from original trial reports using a specifically designed data extraction form containing information on study characteristics (first author, publication year, ClinicalTrials.gov Identifier code, study design, follow-up duration), patient characteristics (inclusion criteria, mean age, percentage of genetic features), sample sizes and the details of interventions with comparisons and outcomes. We evaluated hazard ratios (HRs) with standard errors of PFS and OS between different treatments. To minimize data entry error, all data were entered in duplicate and cross-checked for accuracy, and disparities would be discussed by a group meeting.

### 2.6. Data Synthesis and Statistical Analysis

A network meta-analysis was conducted to compare the treatment outcomes between several novel targeted agents. A network meta-analysis combines direct and indirect estimates of relative treatment effects in one analysis [22]. The network meta-analysis was performed using the frequentist model in the statistical package ‘netmeta’ 0.9-0 (https://cran.r-project.org/web/packages/netmeta/index.html) in R 3.4.2 (R Core Team, Vienna, Austria) [23] and was reported according to the PRISMA extension statement for network meta-analyses. We used the conventional regimen with Ofatumumab as the reference treatment, a treatment approved by the FDA of the US for the treatment of CLL in patients who refractory to fludarabine-based and alemtuzumab-based regimens [24]. 

From the preliminary literature search, we found that evidence from RCTs forms two separate networks. To connect these networks, we included an observational study of individualized indirect comparison of the RESONATE and HELIOS trials published by Hillmen et al. [25], which compares ibrutinib (Ibr) to ibrutinib plus bendamustine rituximab (IbrBR) and ibrutinib (Ibr) to bendamustine rituximab (BR). The resulting joint network could then be analyzed using the Lu & Ades model implemented in Netmeta. Nevertheless, this addition of data from the Hillmen study could introduce biases. To minimize such bias, we used the method of design-adjusted analysis [26], a statistical method designed for combining randomized and non-randomized evidence while minimizing the potential biases of non-RCT data. We assume that estimates from a non-randomized study are unbiased (β = 0), with the uncertainty reflected by w. When w = 1, the non-randomized study was taken at face value, when w = 0, the non-randomized study was ignored. Researchers can adjust the value of w to control the amount of confidence placed on a non-randomized study. We conducted design-adjusted analysis [26] and reduced the weight carried by non-randomized studies. In our network meta-analysis, we assigned a smaller weight to the individualized indirect comparison by Hillmen et al. by doubling the variance of its estimate. Hazard risk (HR) with 95% confidence interval (CI) was calculated using the random-effect network meta-analysis (NMA) according to the UK NICE guidelines [27]. *P* < 0.05 was considered statistically significant. 

A network plot was produced to represent the data from all trials included in the analysis [22]. The contribution of each direct comparison to the network estimate was calculated according to the variance of the direct treatment effect and the network structure, later summarized in a contribution plot [28]. A forest plot of the estimated summary effects, along with CIs for all comparisons, summarizes the relative mean effect and prediction on each comparison in one plot [29]. 

We calculated the ranking probabilities of each treatment for each outcome and used surface under the cumulative ranking curve (SUCRA) analysis to summarize the rankings. SUCRA is a simple transformation of the mean rank that provides a hierarchy of the treatments and accounts for the location and variance of all relative treatment effects [30,31,32]. The larger the SUCRA value (i.e., closer to 1), the higher the rank of the intervention. 

## 3. Results

### 3.1. Systematic Literature Review

The initial screening retrieved 684 citations from the databases. After removing duplicates, 540 citations remained. Further screening using the title or abstract to meet the clinical trial requirement excluded 449 studies. In the next phase, 91 full texts were assessed for potential eligibility, which excluded 83 studies for a variety of reasons (e.g., non-novel targeted agents treatment, non-relapse or refractory CLL). The remaining eight studies included one indirect comparison article and seven RCTs. All RCTs were high-quality, phase III trials reported as complete research articles, all were included in our quantitative synthesis. Figure 1 shows the PRISMA flowchart.

The seven included studies investigated the following treatment options: (1) Ibrutinib (Ibr), (2) ibrutinib plus bendamustine rituximab (IbrBR), (3) venetoclax rituximab (VR), (4) idelalisib plus ofatumumab (IdeOfa), (5) idelalisib plus bendamustine rituximab (IdeBR), (6) duvelisib (Duv), (7) bendamustine rituximab (BR), (8) rituximab (R), and (9) ofatumumab (Ofa).

Table 1 summarizes the characteristics of the included trials. The mean age ranged from 63 to 69 years, and the time from the initial diagnosis to randomization into the trial ranged from 58.1 to 93.6 months. Patients received a median of two (range two to three) previous lines of treatment. The mean percentage of Rai stage >III in each trial ranged from 18% to 63.7%, while the fraction of patients with del(17p) mutation ranged from 0% to 40%. The median follow-up time ranged from 11 to 23.8 months. The overall response rate (ORR) was greater for single Ibr and VR treatments than for the other treatments (ORR, >90%) Table 2.

### 3.2. Data Extraction

The seven included RCTs reported the HRs of the PFS and OS together with corresponding confident intervals. Additionally, we included the adjusted indirect comparison data published by Hillmen et al. [25]. We extracted the HRs of the PFS and OS with the corresponding confident intervals from the indirect comparison to complete the integrated NMA Table 2.

### 3.3. Network Meta-Analysis

A total of 2512 patients were included in our NMA. Figure 2 presents the integrated network scheme for R/R CLL. However, because the number of included studies was only seven, too few to yield a robust estimate of random effect, we, therefore, undertook fixed effect network meta-analysis. 

### 3.4. Clinical Efficacy (PFS)

Figure 3 presents the results of NMA for PFS, in which Ofa was used as the comparator. All treatments were sorted based on their ranking and accompanied by the HR with the 95% CI versus Ofa. With the exception of conventional BR and R, all newly developed novel-targeted-agent–based therapies were significantly more effective than Ofa and reduced the risk of progression or death by more than 48%. In the analysis of overall PFS, Ibr and VR were more effective than the other treatments for patients with R/R CLL. Both of these treatments resulted in a more favorable HR than did Ofa (Ibr: HR, 0.10; 95% CI, 0.07–0.14; VR: HR, 0.10; 95% CI, 0.05–0.21). In other words, treatments with Ibr or VR reduced the risk of disease progression or death by 90% compared to conventional Ofa. Furthermore, the SUCRA value of Ibr was 0.92, showing that the cumulative ranking probability of being the best treatment option in the network analysis was 92%. VR had an approximate SUCRA value of 0.90, a 90% chance of being the best treatment option.

### 3.5. Clinical Efficacy (OS)

Figure 4 presents the results of the NMA for OS, in which ofatumumab was used as the comparator. Only VR (HR, 0.335; 95% CI, 0.112–0.997) and Ibr (HR, 0.361; 95% CI, 0.208–0.627) were significantly more effective than the comparator. VR and Ibr were ranked as the most effective treatments, with similar SUCRA values of 0.85 and 0.84, respectively. The other treatments did not significantly differ from that of Ofa but had a trend toward a greater effectiveness of novel targeted agents, with the median HR ranging from 0.335 to 0.99. However, the median OS among most of the included trials had not yet been reached. 

### 3.6. Clinical Efficacy (PFS in Patients without del(17p))

The del(17p) mutation is associated with shorter PFS and resistance to fludarabine-based regimens [2]. According to the National Comprehensive Cancer Network (NCCN) guidelines, the treatment options for CLL depended on whether a patient carries del(17p) [33]. Therefore, analysis of the results of patients without del(17p) as a subgroup is important.

All of the seven RCTs reported HRs of PFS in del(17p) patients (total, 2021 patients). Results of the NMA of PFS in del(17p) patients are shown in Figure 5. Six of nine treatments resulted in significantly higher PFS than did Ofa. Ibr was ranked as the most effective treatment (HR, 0.09; 95% CI, 0.06–0.15), with a SUCRA value of 0.92. VR treatment was the second-best option based on the SUCRA value (HR, 0.10; 95% CI, 0.05–0.23; SUCRA, 0.87). However, the HR versus Ofa was similar between VR and Ibr in this subgroup analysis of patients with del(17p). This result is consistent with that of the PFS analysis.

### 3.7. Results of Efficacy Outcome Synthesis

We incorporated the evidence, resulting from our analysis, of the relative treatment effects of PFS and OS. Details of the head-to-head comparison of outcomes are shown in Appendix A. The two-dimensional graphs are shown as a scatter plot in Figure 6, illustrating the probability of the best efficacy outcome of PFS and OS for all comparisons. Data are reported by SUCRA values determined by our NMA. Ibrutinib and VR were the most effective treatment options based on PFS and OS.

## 4. Discussion

This systematic review and NMA compares the efficacy of several new novel targeted agents for the treatment of CLL. Our results show that Ibr and VR are the most efficacious agents with respect to PFS and OS in patients with R/R CLL. According to our NMA, treatment with each of the novel targeted agents resulted in greater PFS than did traditional R, Ofa, or BR regimens. Ibr and VR were ranked as the best treatment options based on PFS analysis. We also conducted subgroup analysis in patients without del(17p) mutation. On account of the HELIOS trial populations excluded patients carrying del(17p), differences between the patient populations should be considered to produce more accurate results. In the analysis of OS, both Ibr and VR prolonged the elapsed time from trial randomization to death. Thus, both Ibr and VR treatment result in greater PFS and OS, supporting the recommendations made by the NCCN identifying Ibr and VR as the preferred agents for patients with R/R CLL, especially those with del(17p) [34].

Our results are consistent with a recent analysis that indirectly compared four treatments: ibrutinib, idelalisib plus ofatumumab, ofatumumab and physician’s choice. A strong and consistent trend of superior results for ibrutinib treatment was observed. The analysis demonstrated that ibrutinib treatment resulted in a higher ORR and longer PFS and OS as compared to idelalisib plus ofatumumab, ofatumumab monotherapy, and conventional chemotherapy. However, VR was not included in this study.

Our study used HR as the measure of survival and OS as the secondary endpoint. The network included the results of more recent trials and the individualized indirect comparison reported by Hillmen et al. [25] synthesized into one single network for R/R CLL. Using a frequentist model NMA, it was possible to combine evidence from seven phase III RCTs, including nine different treatments, into one NMA. Therefore, the present results provide important information for clinical physicians to use in making treatment decisions for patients with R/R CLL.

Our NMA results indicate that Ibr monotherapy is more beneficial than Ibr combination therapy for treating patients with R/R CLL. Similarly, Burger et al. [35] reported no improvement in treatment outcomes offered by the addition of rituximab to the ibrutinib regimen in treatment-naïve (*n* = 27) or R/R CLL (*n* = 181) patients. The addition of rituximab did not improve the ORR (Ibr, 92.3%; Ibr + R, 92.3%) or the two-year PFS (Ibr, 95%; Ibr + R, 92.5%). Furthermore, the individualized indirect comparison by Hillmen et al. [25] suggested the superiority of single-agent ibrutinib to ibrutinib plus BR for PFS and OS in patients with R/R CLL, showing that the benefit of ibrutinib plus BR mainly reflected the effect of ibrutinib. According to currently available evidence, the addition of conventional agents (bendamustine, rituximab) did not improve the efficacy but increased greater toxicities. However, there is no proper study directly compared ibrutinib with ibrutinib plus BR. On the basis of current evidence, Ibr monotherapy should remain the standard of care. 

While all of the studies included in our NMA were clinical trials, they were conducted in selected populations and in different locations. To gain insight into practice patterns, real-world evidence should be taken into consideration. Ibrutinib demonstrates long-term efficacy and tolerability not only in these trials but also in actual practice. In a 30-month follow-up real-world cohort study reported by Winqvist et al. [36], ibrutinib treatment resulted in an ORR of 84%, PFS of 52% and OS of 63% at a median follow-up of 30 months. The study cohort included 95 patients with a median age of 69 years (del(17p)/TP53 mutation, 63%, the median number of previous therapies, 3). Another five-year study of single-agent Ibr treatment for patients with R/R CLL was reported by Brien et al. [37]. This study included 101 R/R CLL patients with a median age of 68 years; del(17p), 34%; median number of previous therapies, four; and median follow-up time of 39 months. The ORR was 86% and the median PFS was 52 months. Thus, these real-world studies of single-agent ibrutinib treatment show a durable response in patients with R/R CLL, supporting our NMA results showing that mono-Ibrutinib therapy is the most favourable option for reducing both the risk of disease progression and death in R/R CLL patients.

While our results indicate that Ibr is the first choice of novel targeted agents. The results of the MURANO trial [10] of venetoclax and rituximab may challenge this conclusion. Evidence from real-world studies of venetoclax remains quite sparse. A prospective study by Mato et al. [38] investigated the efficacy of novel targeted agents in 683 CLL patients with no previous kinase inhibitor (idelalisib or ibrutinib) treatment. The median number of previous therapies was two, and approximately one-third of the patients had high-risk genetic features identified as del17p, del11q, or complex karyotype. With a median follow-up of 17 months, R/R CLL patients receiving Ibr-based therapy experienced a significantly greater PFS as compared to idelalisib-based therapy (HR, 2.8; CI, 1.9–4.1; *P* < 0.001). Subsequent regimens after initial kinase inhibitor failure were classified as kinase inhibitor-based therapy (ORR: 58.5%), venetoclax (ORR: 73.6%) and chemoimmunotherapy combinations (ORR: 49.9%). The authors concluded that Ibr is superior to idelalisib for treating patients with R/R CLL. The efficacy of venetoclax was established in patients in whom previous kinase inhibitor treatment (idelalisib or ibrutinib) failed. Another real-world retrospective cohort analysis [39] of patients with CLL (98% R/R) demonstrated that venetoclax has a similar ORR (72.1%) to that previously reported in a clinical trial [40]. In addition, as the authors concluded, venetoclax was effective in patients who had previous ibrutinib therapy and carried mutations known to confer resistance to ibrutinib.

In current real-world practice, ibrutinib monotherapy is effective and provides better survival outcomes than other treatments, suggesting its use as the standard of care for R/R CLL patients. Although real-world experience with VR is limited, venetoclax monotherapy demonstrated promising efficacy and a high response rate. However, optimization of novel targeted agents is required for their further evaluation.

This study has several limitations. First, because of the limited number of studies included in our NMA, the connections within our network are sparse. The treatment comparisons were therefore estimated by either direct or indirect evidence. Consequently, it is not feasible to evaluate inconsistencies in treatment effects between direct and indirect comparisons [41]. Furthermore, our fixed effect models may underestimate the uncertainty of treatment effects. Second, to provide missing links between treatments in our integrated network, we combined the data from an individualized, indirect comparison of the RESONATE [5] and HELIOS [6] trials reported by Hillmen [25]. In this indirect comparison, the author used patient-level data from both studies. The biases in cross-trial comparisons had already been adjusted, including treatment and clinically relevant prognostic variables as covariates (age, gender, Rai staging, ECOG score, del11q status, refractory status, number of prior lines of therapy, bulky disease, IgVH status). However, including such data in an NMA may increase the risk of producing a biased effect estimate and influence the results. Third, adverse-event profiles are not accounted for in our NMA. Ibrutinib displayed a manageable safety profile in the results of previous RCTs [3,8]. However, insight from real-world practice, adverse event profiles and treatment discontinuation were emphasized. 

Toxicity was the most common reason for ibrutinib discontinuation in CLL patients, approaching 50% [42]. Atrial fibrillation, infectious complications, and cytopenias were the most commonly described adverse events in patients treated with ibrutinib [43]. In addition, an uncommon but potentially life-threatening complication, tumor lysis syndrome, has been recently reported in venetoclax administration [44]. However, a comparison of safety issues or reasons for treatment discontinuation was not included in our NMA because of insufficient data.

Despite these limitations, our NMA provides insight into the rank order of treatment efficacy according to the presently available evidence. This information may help doctors to select the most appropriate treatment for each individual patient, in the absence of direct head-to-head comparison trials. The findings of this NMA suggest that ibrutinib monotherapy and venetoclax rituximab (VR) are most likely to become the most effective treatments with respect to long-term PFS and OS for patients with R/R CLL.

## 5. Conclusions 

According to our NMA, treatment with each of the novel targeted agents resulted in greater PFS than did traditional R, Ofa, or BR regimens. Both ibrutinib monotherapy and venetoclax rituximab have a high probability of being the most effective treatments for a relapse or refractory chronic lymphocytic leukemia with respect to long-term progression-free survival and overall survival.

## Figures and Tables

**Figure 1 jcm-08-00737-f001:**
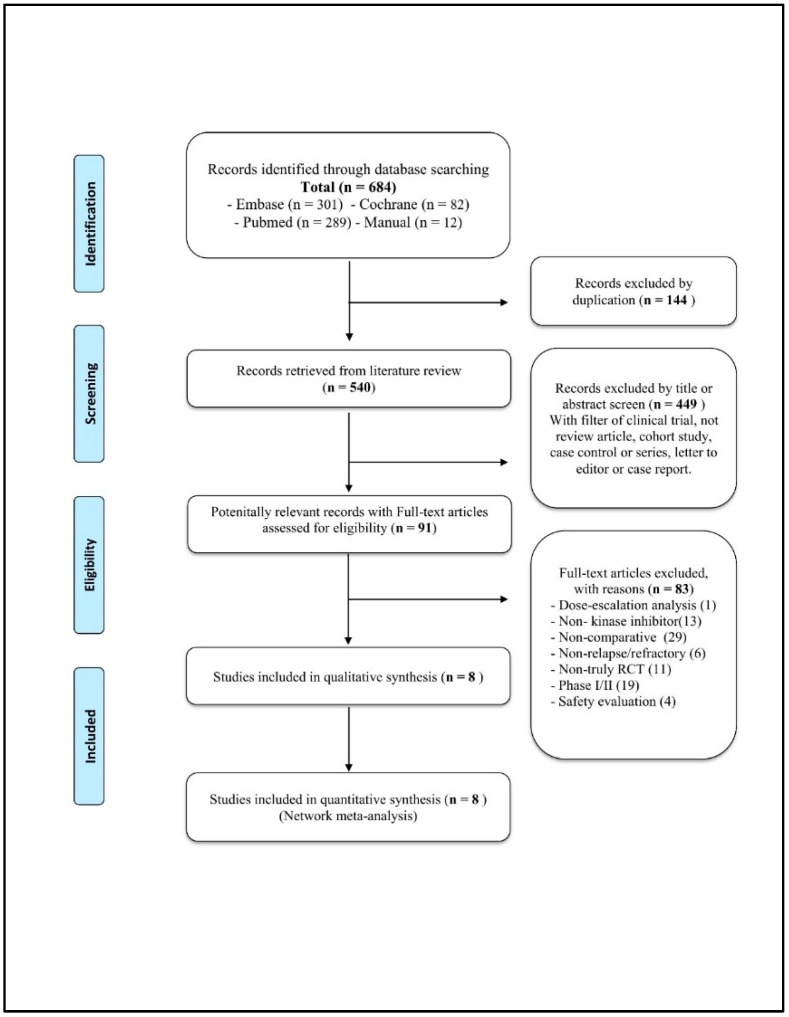
Preferred Reporting Items for Systematic Reviews and Meta-Analyses (PRISMA) flowchart of refractory/relapse chronic lymphocytic leukemia.

**Figure 2 jcm-08-00737-f002:**
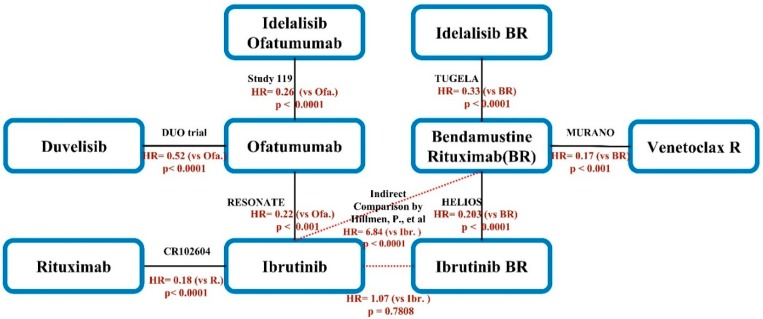
Schematic diagram of the network of evidence used in network meta-analysis (NMA). The black line indicates a direct comparison from RCT. The name of the trial and the published year are noted in black. The hazard ratio of PFS is noted in orange. The red dotted line indicates the individualized indirect comparison data published by Hillmen et al. HR: Hazard ratio, BR: Bendamustine + Rituximab, R: Rituximab, P: P-value.

**Figure 3 jcm-08-00737-f003:**
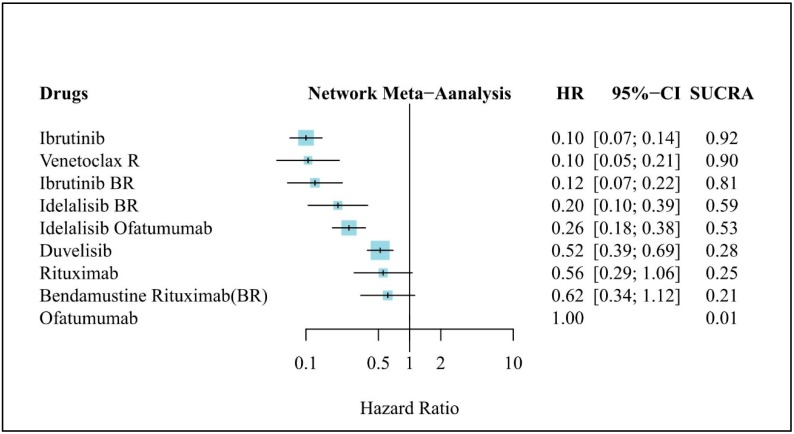
Network meta-analysis results of treatment efficacy in refractory/relapse (R/R) chronic lymphocytic leukemia (CLL): Forest plot of PFS in R/R CLL. HR: Hazard ratio; CI: Confidence interval, SUCRA: Surface under the cumulative ranking curve, BR: Bendamustine + Rituximab, R: Rituximab.

**Figure 4 jcm-08-00737-f004:**
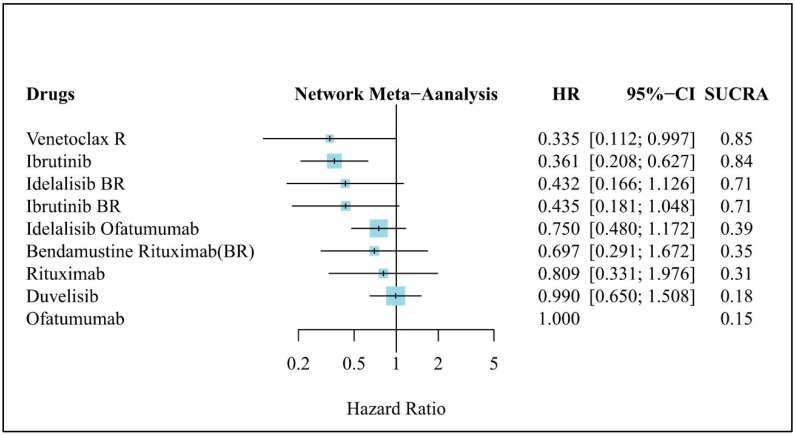
Network meta-analysis results of treatment efficacy in refractory/relapse (R/R) chronic lymphocytic leukemia (CLL): Forest plot of OS in R/R CLL. HR: Hazard ratio, CI: Confidence interval, SUCRA: Surface under the cumulative ranking curve, BR: Bendamustine + Rituximab, R: Rituximab.

**Figure 5 jcm-08-00737-f005:**
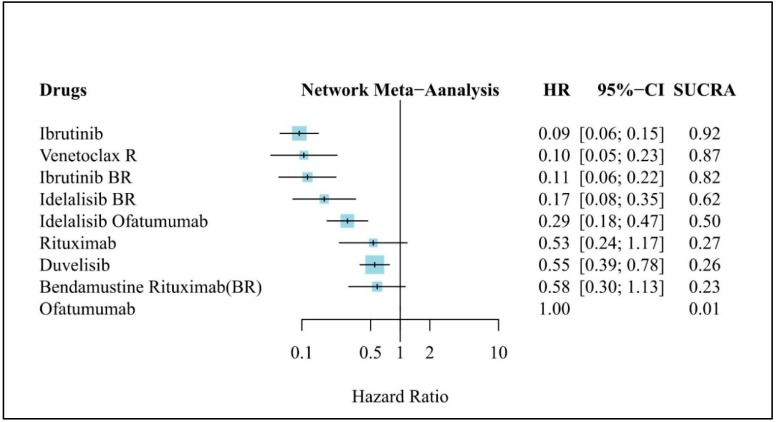
Network meta-analysis results of treatment efficacy in refractory/relapse (R/R) chronic lymphocytic leukemia (CLL): Forest plot of PFS in R/R CLL patients without del(17p) mutation. HR: Hazard ratio, CI: Confidence interval, SUCRA: Surface under the cumulative ranking curve, BR: Bendamustine + Rituximab, R: Rituximab.

**Figure 6 jcm-08-00737-f006:**
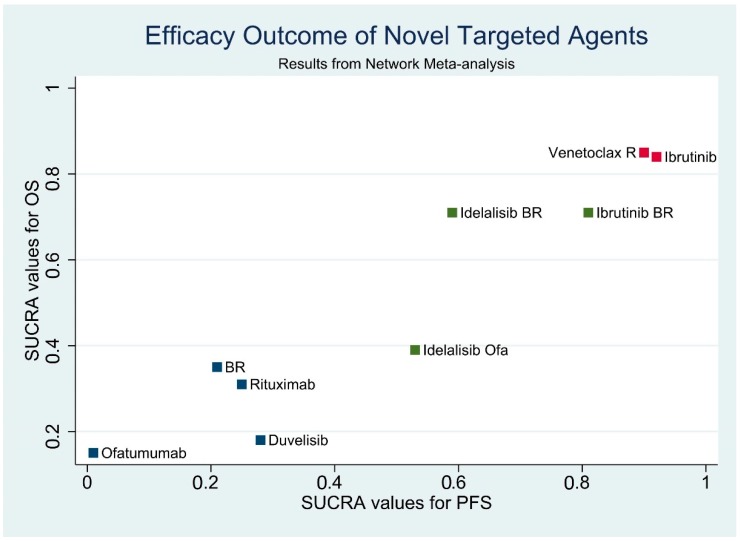
Efficacy outcomes for progression-free survival (PFS) and overall survival (OS) in network meta-analysis: Scatter plot. Data are reported by SUCRA values from the results of our network meta-analysis. X-axis, SUCRA values for PFS. Y-axis, SUCRA values for OS. Red nodes indicate the most effective treatment options. Green nodes indicate treatment options more effective than conventional regimens. Navy blue nodes indicate treatment options as effective as conventional regimens. PFS: Progression-free survival, OS: Overall survival, B: Bendamustine, R: Rituximab, Ofa: Ofatumumab, SUCRA: Surface under the cumulative ranking curve.

**Table 1 jcm-08-00737-t001:** Trial details and patient characteristics.

Trial Name NCT Number	First Author Published Year	Trial Design	Experimental Arm	Control Arm	Mean Age	Rai Stage > III (%)	Number of Prior Therapies	Del(17p) Mutation (%)
Study 119 (NCT01659021)	Jones, J.A. 2017	OP, RCT Phase III	Idelalisib Ofatumumab ^a^ (Ide: 150 mg bid po)	Ofatumumab ^b^	67.7	63.70%	3 vs. 3	40% vs. 38%
DUO trial (NCT02004522)	Flinn, I.W. 2018	OP, RCT Phase III	Duvelisib (25 mg bid po)	Ofatumumab ^b^	69	56%	2 vs. 2	21% vs. 28%
RESONATE (NCT01578707)	Brown, J.R. 2018	OP, RCT Phase III	Ibrutinib (420 mg qd po)	Ofatumumab ^b^	66.8	57.30%	3 vs. 2	32% vs. 33%
CR102604 (NCT01973387)	Huang, X. 2018	OP, RCT Phase III	Ibrutinib (420 mg qd po)	Rituximab ^c^	63.6	77.70%	2 vs. 2	27.1% vs. 24.1%
HELIOS (NCT01611090)	Chanan-Khan, A. 2016	DB, RCT Phase III	Ibrutinib BR ^d^ (Ibr 420 mg qd po)	BR ^d^	63.5	42.50%	2 vs. 2	0% vs. 0%
MURANO (NCT02005471)	Seymour, J.F. 2018	OP, RCT Phase III	Venetoclax R ^e^	BR ^d^	65.3	18%	2 vs. 2	23.7% vs. 23.6%
TUGELA (NCT01569295)	Zelenetz, A.D. 2017	DB, RCT Phase III	Idelalisib BR ^d^ (Ide.: 150 mg bid po)	BR ^d^	63	45.50%	2 vs. 2	18% vs. 19%

DB: Double blind, OP: Open label, RCT: Randomized control trial, B: Bendamustine, R: Rituximab, Ide: Idelalisib, Ibr: Ibrutinib. ^a^ Ofatumumab for a total of 12 infusions (300 mg on Day 1, followed by 1000 mg weekly for seven weeks, and then 1000 mg every four weeks for four doses). ^b^ Ofatumumab for a total of 12 infusions (300 mg on Day 1, followed by 2000 mg weekly for seven weeks, and then 2000 mg every four weeks for four doses). ^c^ Rituximab: Up to six cycles (total of eight doses administered by intravenous infusion) 375 mg/m^2^ on Day 1 of Cycle 1, 500 mg/m^2^ on Day 15 of Cycle 1 (Weeks 1–4), 500 mg/m^2^ on Day 1 and Day 15 of Cycle 2 (Weeks 5–8), and 500 mg/m^2^ on Day 1 of Cycles 3–6 (Weeks 9–24). ^d^ Bendamustine 70 mg/mg^2^/day on two consecutive days every 28 days administered intravenously for a maximum of 12 infusions. Rituximab 375 mg/m^2^ on Day 1, then 500 mg/m^2^ every 28 days administered intravenously for a maximum of six infusions. ^e^ Venetoclax was administered at an initial dose of 20 mg via tablet orally QD, incremented weekly up to a maximum dose of 400 mg during a five-week ramp-up period. Venetoclax will be continued at 400 mg QD from Week 6 (Day 1 of Cycle 1 of combination therapy) onwards up to disease progression (PD) or two years. Rituximab 375 mg/m^2^ on Day 1, then 500 mg/m^2^ every 28 days administered intravenously for a maximum of six infusions.

**Table 2 jcm-08-00737-t002:** Extracted data of included randomized controlled trials.

Trial Name/ID Number	Experimental Arm	Control Arm	Patient Numbers	Median Follow-Up	Overall Response Rate	Median PFS	Hazard Ratio (95% CI), PFS	Median OS	Hazard Ratio (95% CI), OS
Study 119	Idelalisib Ofatumumab	Ofatumumab	174 vs. 87	16.1 vs. 5.8 months	75.3% vs. 18.4%	16.4 vs. 8.0 months	0.26 (0.18 to 0.38)	NR vs NR	0.750 (0.48 to 1.18)
DUO trial	Duvelisib	Ofatumumab	160 vs. 159	22.4 months	73.8% vs. 45.3%	13.3 vs. 9.9 months	0.52 (0.39 to 0.7)	NR vs NR	0.99 (0.65 to 1.5)
RESONATE	Ibrutinib	Ofatumumab	195 vs. 196	19 months	90% vs. 25%	NR vs. 8.1 months	0.10 (0.07 to 0.15)	NR vs NR	0.361 (0.208 to 0.628)
CR102604	Ibrutinib	Rituximab	106 vs. 54	17.8 months	67.9% vs. 7.4%	NR vs. 8.3 months	0.18 (0.105 to 0.308)	NR vs 26.1 months	0.446 (0.221 to 0.9)
HELIOS	Ibrutinib BR	BR	289 vs. 289	17 months	86% vs. 69%	NR vs. 13.3 months	0.203 (0.15 to 0.276)	NR vs NR	0.628 (0.385 to 1.024)
MURANO	Venetoclax R	BR	194 vs. 195	23.8 months	93.3% vs. 67.7%	NR vs. 17 months	0.17 (0.12 to 0.26)	NR vs NR	0.48 (0.25 to 0.9)
TUGELA	Idelalisib BR	BR	207 vs. 209	14 months	70% vs. 45%	20.8 vs. 11.1 months	0.33 (0.25 to 0.44)	NR vs 31.6 months	0.62 (0.42 to 0.92)

PFS: Progression-free survival, OS: Overall survival, B: Bendamustine, R: Rituximab, NR: Not reached, CI: Confidence interval.

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
