# Peer review of "Treatment Outcomes of Novel Targeted Agents in Relapse/Refractory Chronic Lymphocytic Leukemia: A Systematic Review and Network Meta-Analysis"

_jcm, 2019, doi:10.3390/jcm8050737_

Reviewer 1 Report

The manuscript by Po-Huang Chen et al describes the current trends in Chronic Lymphocytic Leukemia (CLL) treatment. In particular, they performed a network meta-analysis to assess the relative efficacy of novel kinase inhibitors for the treatment of Relapse or Refractory Chronic Lymphocytic Leukemia (R/R CLL). 

After a brief but comprehensive presentation of the conventional therapy for CLL and its drawbacks, a systematic literature search is performed, 7 high-quality studies (phase III randomised controlled trials) are selected and data were analysed and synthesized using network meta-analysis. The inclusion of the specific ClinicalTrials.gov Identifier helps to redirect the reader immediately to related information.The manuscript includes more than 40 references and reflects the state of the art in this area of research. The manuscript is convincing, and it offers new suggestions in an important scientific field. It will certainly be an important tool for researchers working in the R/R CLL for therapeutic purposes and could provide treatment recommendations to physicians in daily clinical practice. Although the manuscript has been carefully prepared and English language and style are fine, there are some mistakes.  

I would like to strongly recommend some points to further improve the manuscript:

·      ThisThe network meta-analysis assesses the relative efficacy of novel kinase inhibitors for the treatment of relapse or refractory chronic lymphocytic leukemia. 

·      All kinase inhibitor therapies were significantly more effective than ofatumumab and demonstrated promising prolongation of progression free survival (PFS)….

·      The results of clinical trials evaluating progression free survival (PFS) and OS of small-molecule inhibitor 

·      If figure is a word of the sentence, the author should delate the brackets. See for instance

 (Figure 1shows the PRISMA flowchart.

(Table 1)summarizes the characteristics of the included trials.

·      The other treatments did not significantly differ differentfrom that of Ofa 

·      As a resultTherefore, analysis of the result of patients without del(17p) as a subgroup is important.

·      We incorporated the certainty ofevidence,

Author Response

Response:

Thanks for your valuable comments with our sincerest appreciation. We agree with your comments completely. We revised all the mistakes of manuscripts according to your suggestions.

Reviewer 2 Report

The manuscript title “Treatment Outcomes of Novel Kinase Inhibitors in Relapse/Refractory Chronic Lymphocytic Leukemia: A Systematic Review and Network Meta-analysis” by Po-Huang Chen, Ching-Liang Ho, Chin Lin, Yi-Ying Wu, Tzu-Chuan Huang, Yu-Kang Tu and Cho-Hao Lee M.D is focus on design a new network meta-analysis to assesses the relative efficacy of novel kinase inhibitors for the treatment of relapse or refractory (R/R) chronic lymphocytic leukemia (CLL) patients. The authors conclude that ibrutinib monotherapy and venetoclax-rituximab have the high probability of being the most effective treatment for relapse and refractory in chronic lymphocytic leukemia with respect to long term progression free survival and overall survival. 

Mayor concerns

The authors propose that ibrutinib monotherapy and venetoclax-rituximab as the most effective treatment for R/R in CLL but even though they suggest that ibrutinib as the first choice, is not clear why is not venetoclax-rituximab. In my opinion it would be interesting to discuss it better in the text. What about the toxicity or any kind of life threating complications more frequent in one than the other treatment.  

Minor comments

1-     In the figure 1 flow chart is mention 8 studies were included but, in the text, said only 7 were included. 

2-     In the figure 2, the orange values (HR and pvalue) are not clear. Please, try to choose other color or at least make it bigger.

3-     In the discussion section:

- 2ndparagraph_ ‘Our results are consistent with a recent analysis that indirectly compared 3 studies: the RESONATE study, Study OMB114242 and Study 119’.

I do not understand how the OMB114242 study is consistent to the results. The OMB114242 study compare ofatumumab versus physicians’ choice treatment in patients with bulky fludarabine-refractory CLL and explore extended versus standard-length ofatumumab treatment. The results showed that the median PFS was significantly longer for ofatumumab versus physicians’ choice. Because of that I don’t understand the comment. 

- 6paragraph_ ‘A prospective study by Matoet al.’ Add space Mato el al 

Author Response

Major concerns

The authors propose that ibrutinib monotherapy and venetoclax-rituximab as the most effective treatment for R/R in CLL but even though they suggest that ibrutinib as the first choice, is not clear why is not venetoclax-rituximab. In my opinion it would be interesting to discuss it better in the text. What about the toxicity or any kind of life threating complications more frequent in one than the other treatment.

Response:

Many thanks for your valuable comments on our manuscript. We completely agree with you that when applying an intervention to patients, both the benefit and harm should be taken into consideration. However, because of the lack of sufficient evidence, the risk of life-threatening complications caused by these drugs cannot be analyzed in our network meta-analysis.

As a result, we could only compare treatment efficacy and our NMA showed that ibrutinib monotherapy and venetoclax rituximab (VR) seemed to be the most effective treatments with respect to long-term progress free survival (PFS) and overall survival (OS) for patients with R/R CLL.

Current evidence on the potential toxicity of venetoclax-rituximab was limited; there was only one large RCT (MURANO study) and no large cohort study or real-world study has been reported. We could not compare the relative risk of serious adverse events of ibrutinib with that of venetoclax-rituximab.

Ibrutinib was approved by FDA since 2013, a lot of surveillance and real-world data had reported the short-term and long-term side effects, most of which were manageable. Venetoclax is a novel agent (approved by FDA since 2018 for CLL/SLL). While its promising effect was showed, its short-term and long-term side effects remain unclear. Therefore, in the absence of head-to-head studies and taking into consideration the limitations of the indirect comparison, we made a conservative conclusion in our study. It remains essential to conduct further phase III RCTs to obtain direct head-to-head evidence.

Minor comments

1-     In the figure 1 flow chart is mention 8 studies were included but, in the text, said only 7 were included.

Response:

There were 8 studies included in our study, including 7 phase III RCT and 1 indirect comparison article. We have revised the text and apologized for not being clear.

The revised text now reads:

The remaining 8 studies included 1 indirect comparison article and 7 RCTs. All RCTs were high-quality, phase III trials reported as complete research articles; all were included in our quantitative synthesis.

2-     In the figure 2, the orange values (HR and p-value) are not clear. Please, try to choose other color or at least make it bigger.

Response:

Many thanks for your suggestion. We have changed the color of values (HR and p-value) to dark red and used a larger font size in figure 2.

3-     In the discussion section:

- 2ndparagraph_ ‘Our results are consistent with a recent analysis that indirectly compared 3 studies: the RESONATE study, Study OMB114242 and Study 119’.

I do not understand how the OMB114242 study is consistent to the results. The OMB114242 study compare ofatumumab versus physicians’ choice treatment in patients with bulky fludarabine-refractory CLL and explore extended versus standard-length ofatumumab treatment. The results showed that the median PFS was significantly longer for ofatumumab versus physicians’ choice. Because of that I don’t understand the comment.

Response:

Many thanks for your valuable comment. In the discussion section, we discussed a recent analysis [34] which indirectly compared three studies including the RESONATE study, Study OMB114242 and Study 119. In the analysis, the result of Study OMB114242 was a bridge, and ofatumumab was the common comparator to connect the indirect comparison. The conclusion of the indirect meta-analysis[34] showed that ibrutinib appears to have a higher overall response rate (ORR), longer PFS and OS compared to idelalisib plus ofatumumab and to physician's choice.

We indicated that the result of the analysis[34] is similar to ours. Mono-ibrutinib appears to have a superior treatment effect versus other combinations. We have revised the sentence in our manuscript as follows:“Our results are consistent with a recent analysis that indirectly compared 4 treatments: ibrutinib, idelalisib plus ofatumumab, ofatumumab and physician's choice [34].”

- 6paragraph_ ‘A prospective study by Matoet al.’ Add space Mato el al

Response:

We have corrected the error in our text.
